# COVID-19 Pandemic: Impediment or Opportunity? Considerations Regarding the Physical-Health Impact and Well-Being among Romanian University Students

**Mihai Adrian Olanescu [1], Marius Adrian Suciu [1,\*] , Claude Scheuer [2] and Miruna Peris [3]**

1    Faculty of Automotive, Mechatronics and Mechanical Engineering, Technical University of Cluj-Napoca, 400641 Cluj-Napoca, Romania
2    Faculty of Humanities, Education and Social Sciences, University of Luxembourg, 1855 Luxembourg, Luxembourg
3    Faculty of Industrial Engineering, Robotics and Production Management, Technical University of Cluj-Napoca, 400641 Cluj-Napoca, Romania
\*    Correspondence: adrian.suciu@mdm.utcluj.ro

**Abstract:** The global COVID-19 pandemic that is ongoing because of the Coronavirus II (SARS-CoV-2) has had until now a great impact on physical education and sports, especially due to the closure of training facilities and people's lack of motivation. In Romania, physical activity (PA) and physical education (PE) have also been highly affected, especially among university students. To our knowledge, this is the first study conducted in Romania regarding the influence of the lockdown on the physical health and well-being of university students. This study aims to determine if and how the pandemic affected the university students' mental and physical health in Romania, but also to highlight new strategies and approaches suggested by them through a questionnaire to enhance and motivate their participation in physical activities. Based on a cross-sectional design, a survey designed in two languages—Romanian and English—was completed by the students from the Technical University of Cluj-Napoca and foreign students that were on a mobility study visit through the Erasmus+ program in Romania. The questionnaire was filled out by 836 university students (age = 18–24 years; males: 57.73%). The findings showed that quarantine in Romania led to a significant increase in sedentarism among students and a reduction in energy for physical activity, which also affected their psychological well-being. The absence of social interactions and onsite classes led to a sedentary lifestyle among students and increased their fear and stress levels. The cancellation of all sports events and PA also led to fewer students remaining physically active. Time off training and leisure time activities caused by the ongoing restrictions might be used for new purposes, such as goal setting, overcoming injury, improving mobility, psychological development, and emphasizing strength sports for health. During the COVID-19 lockdown, sedentarism increased dramatically, especially among young people.

**Keywords:** physical activity; physical education; COVID-19; students; lockdown; sedentarism

## 1. Introduction

The consequences of the easy airborne contamination of SARS-CoV-2 have powerfully affected Romania as well as the whole of Europe. Restrictions due to COVID-19 were also a big challenge for amateur and elite sports professionals [1]. Because of the absence of any vaccine or any other pharmaceutical treatment at the beginning of the pandemic, the measures taken against the spread of the virus included population separation, reducing as much as possible human contact, and quarantine—restrictions of movement. The decline in physical activity (PA) should raise a red flag as it is a great concern due to the long-term consequences for healthcare systems and public health [2–7].

COVID-19 first appeared in Romania on 26 February 2020, and the number of confirmed cases until 19 March was 277 [8]. The universities were first closed in March because of the guidelines and recommendations of the government. Movement outside the home was allowed only for specific reasons, while people were more and more afraid of the new virus.

Despite all the health benefits of PA, many people from Romania and worldwide are generally not as physically active as the World Health Organization (WHO) guidance recommends [6,9]. In the WHO PA recommendations, 150 min of physical activity a week or intensive physical activity of 75 min should be performed by adults. According to representative data from Romania based on adults aged 18 to 35 years obtained from the "Eurobarometer on sport and physical activity", 35% of men and 60% of women aged 15–24 years NEVER or seldom exercise or play sport while 75% men and 77% women aged 25–39 years also never or seldom engage in PA or sports [10].

The problem is that even though isolation and restriction of people's free movement or contact may reduce the risk of virus transmission, there is a high possibility that the reduction of physical activity and daily energy expenditure (EE) might be the cause of the development of other diseases. Self-isolation also promotes physical inactivity and sedentarism, which contributes to depression, obesity, anxiety, cardiovascular vulnerability, and bone loss and can have a deleterious psychological impact [5,11,12]. The positive role of PA in improving general health has been described in the literature. Detraining for a long period, as a forced stop because of the COVD-19, can lead to a decline in maximal oxygen consumption, loss of muscle strength, and reduced capacity for endurance [6,13,14].

Among students, as the pandemic continues, psychological stress symptoms are increasing [9] and so is their need for being physically active. Exercise has significant mental health and body benefits, reduces symptoms of depression and anxiety, increases the level of serotonin, and restores cortisol balance [9,15]. There is evidence that indicates that PA is effective in preventing symptoms of mental health disorders and reduces stress levels [6,16–18].

Objectives. The main objective of the research is to identify and analyze the difficulties and differences that arose with the pandemic in the motivation and involvement of students to practice sports. We wanted to identify aspects regarding the practice of sports activities by students, both before and during the pandemic: the frequency of participation and the main reasons that led to the decrease in exercise during the pandemic, favorite physical activities, the most common sports location since the arrival of COVID-19, the partners with whom they did sports, and the degree of involvement of the other family members in the physical activity.

Based on these facts, the purpose of this article was to determine the impact of the COVID-19 pandemic on the university students' mental and physical health during quarantine in Romania, the correlation between PA and the well-being of students, and the importance of remaining physically active by finding different means of doing so. An online questionnaire was applied to determine students' health priorities and how the pandemic affected or improved their health and how inactivity can be prevented in the future.

Hypotheses. The frequency with which students at the Technical University of Cluj-Napoca practice sports during the pandemic differs depending on the degree of difficulty perceived by them in terms of participation in various sports activities after the appearance of the COVID-19 virus. Specifically, we expect students who find it easy to engage in sports activities during the pandemic to do so with a high frequency, while students who find it difficult, to do sports with a low frequency.

We have started our research on the premise that physical activity has a big impact on health, and mental health, emphasizing the importance of being active. Participating in physical activities/different sports can improve motivation and can help students overcome the difficulties and differences that arose with the pandemic.

## 2. Materials and Methods

### 2.1. Research Protocol

#### 2.1.1. Period and Place of the Research

The present study analyzes the way in which the students from the Technical University engaged in sports practice activities during the pandemic, compared to the period before it. The research study was conducted in Cluj-Napoca, from December 2020 until February 2021.

#### 2.1.2. Samples

A total of 836 students, aged between 18 and 24 years, a representative percentage of the total number of students enrolled in the academic year 2020–2021, at one of the nine faculties within the Technical University of Cluj-Napoca, were included in the study. Of these, 482 (57.72%) were boys and 353 (42.27%) were girls. In the academic year 2020–2021, there were 10,504 students enrolled at the Technical University of Cluj-Napoca, of which 7723 (73.5%) were male students and 3281 (26.5%) were female students.

The inclusion criteria consisted of students with busy programs, meaning students who had a very full curricula depending on the faculty they were enrolled in, but students who were less busy were also considered for this sampling. All the students who attended full-time courses at one of the faculties of the Technical University of Cluj-Napoca, regardless of their year of study, age, department, or specialization, were included in this study. The students completed the online Google form questionnaire over a period of 3 months during the COVID-19 emergency in Romania. Due of the possibility of encountering errors because of the unwillingness to provide a valid response according to AAPOR (American Association for Public Opinion Research) written in "Standard Definitions", ineligible cases that contained meaningless data, representing invalid responses, were removed [6,19].

The students were told, before completing the questionnaire, that the survey is totally anonymous, and they could read the following brief description of the study, its aim, and the declarations of anonymity and confidentiality: "This form is intended for all students at the Technical University of Cluj-Napoca. All answers provided in this form will be used to optimize the learning procedures regarding sports during these challenging times. By completing this form, you agree to be part of this study. All the information is confidential. We want to thank you for completing this form!" [6].

#### 2.1.3. Instruments

The method of quantitative research, the technique of administering questionnaires through a sociological survey, was used. A complex online survey, designed by the NCAA Research team, was conducted through the web survey platform Google Forms. For this research, a questionnaire based on questions from several questionnaires, all designed by the NCAA [20], was compiled in order to suit the research topic. Participation in the study was voluntary, and steps were taken to ensure the confidentiality of all the information collected. Because of the fact that the PE classes were held online, the students had to send a paper with a topic given by the professors. That is why the online survey was sent more by e-mail and WhatsApp, but also through official channels (Microsoft Teams) of the Technical University of Cluj-Napoca. The snowball sampling method was used. In order to obtain more answers, we asked the students to send the questionnaire to as many colleagues as possible, with the request to complete it. The needed questions for the study were taken from the NCAA research questionnaire. That is because it has taken many steps to raise awareness regarding students' mental and physical health needs. Questions about their preferred sports locations and partners before and during the pandemic were some of the topics we were interested in for the research. In addition, it helped determine how scared students were during the pandemic given all the restraints imposed.

### 2.1.4. Statistical Processing

The SPSS statistical package (SPSS v25) was used for all analyses. The results considered to be statistically significant were if $p < 0.05$ in all analyses. The adequacy of the logistic regression model Nagelkerke R Square was used for comparisons, the higher the score, the higher the fitness of the predictive model.

Data regarding the practice of physical activities and sports among university students were evaluated and analyzed using measures of the central tendency for quantitative variables (deviation) and frequencies for categorical variables. The relationships between PA (physical activity) and psychological well-being were compared using different graphics and correlations.

### 2.2. Ethics

The study followed the Helsinki declaration and national guidelines concerning the ethical guidelines and legal requirements. The study was approved by the ethics committee of the University, and each student participated voluntarily.

### 3. Results

University students' physical activity levels decreased, and sedentary activity increased significantly during COVID-19. During COVID-19, students did not engage in sufficient physical activity to offset the increased sedentary behavior.

According to the applied questionnaire, which was meant to find students' involvement in PA before and during the pandemic, we found the following results.

Statistical modeling of the probability that students considered it more difficult to practice sports activities during the pandemic, depending on the reasons invoked by them, with the help of logistic regression, is presented in Table 1.

**Table 1.** The results of the logistic regression regarding the statistical determinants of the probability that it was more difficult for students to practice sports during the pandemic.

| Dependent Variables | Sig. | Exp (B) |
|---|---|---|
| Insufficient indoor space for practicing sports | 0.000 | 0.496 |
| The insecurity of doing sports in public spaces | 0.001 | 0.491 |
| Lack of access to sports facilities | 0.001 | 0.522 |
| Lack of sports equipment | 0.433 | 0.812 |
| Fear of exposure to COVID-19 | 0.043 | 0.615 |
| Lack of training partners | 0.473 | 1.187 |
| Local travel restrictions | 0.008 | 0.601 |
| Closure of sports venues | 0.380 | 0.837 |
| The uncertainty that social distancing is respected | 0.759 | 1.113 |
| Lack of motivation to exercise | 0.001 | 0.548 |
| Lack of time for sports activities | 0.195 | 0.758 |
| I'm not sure how to do the exercises at home | 0.106 | 0.516 |
| Personal and family responsibilities | 0.002 | 2.583 |
| Constant | 0.000 | 22.635 |

The results of the logistic regression obtained following the analysis of the dependent variables presented in the table above show that family and/or personal responsibilities was the most important predictor for students who considered that during the pandemic it was more difficult to practice sports activities, because exp (b) = 2.583. Therefore, it is two and a half times more likely that students who found it more difficult to play sports during the pandemic would be the ones who invoked these reasons.

Two other predictors were also important: lack of training partners, because exp (b) = 1.187, and fear that the social distance rules are not respected, because exp (b) = 1.113.

The following predictors: the insecurity of doing sports in public spaces, lack of access to sports facilities, and lack of motivation to exercise, even if they have a statistically

significant value because of the value of Sig. < 0.005, proved to be less important because Exp (B) has a small value, less than 1.

The adequacy of the logistic regression model is acceptable (Nagelkerke R Square = 0.205), as presented in Table 2.

**Table 2.** The results of the logistic regression regarding the statistical determinants of the probability that students would practice less sports activities during the pandemic, compared to the period before it.

| Dependent Variables | Sig. | Exp (B) |
|---|---|---|
| Insufficient indoor space for practicing sports | 0.000 | 2.080 |
| The insecurity of doing sports in public spaces | 0.004 | 1.862 |
| Lack of access to sports facilities | 0.002 | 1.811 |
| Lack of sports equipment | 0.065 | 0.621 |
| Fear of exposure to COVID-19 | 0.077 | 1.526 |
| Lack of training partners | 0.161 | 1.423 |
| Local travel restrictions | 0.053 | 1.453 |
| Closure of sports venues | 0.062 | 1.470 |
| The uncertainty that social distancing is respected | 0.379 | 0.738 |
| Lack of motivation to exercise | 0.000 | 1.940 |
| Lack of time for sports activities | 0.002 | 1.983 |
| I'm not sure how to do the exercises at home | 0.041 | 2.511 |
| Personal and family responsibilities | 0.113 | 0.620 |
| Constant | 0.000 | 0.022 |

The most important predictors were insufficient interior space for sports, because exp (b) = 2.080, insecurity of going to public spaces for sports, because exp (b) = 1.862, lack of access to different sports locations, because exp (b) = 1.811, lack of motivation to do sports, because exp (b) = 1.940, and lack of time for sports activities, because exp (b) = 1.983.

The following predictors: lack of sports equipment, fear of exposure to COVID-19, lack of training partners, local travel restrictions, closure of sports venues, and I'm not sure how to do the exercises at home, are not statistically significant because of the value of Sig. >0.005, even if the value of Exp (B) is higher than 1.

The adequacy of the logistic regression model is acceptable (Nagelkerke R Square = 0.212), as shown in Table 3.

**Table 3.** The results of the logistic regression regarding the statistical determinants of the probability that students would find it more difficult to practice sports during the pandemic.

| Dependent Variables | Sig. | Exp (B) |
|---|---|---|
| **Physical activities performed before the pandemic** | | |
| A walk lasting at least 10 min | 0.535 | 0.879 |
| Cycling | 0.967 | 1.010 |
| Running or jogging | 0.035 | 1.602 |
| Home activity, fitness, or exercise | 0.688 | 1.089 |
| Physical activities performed online at home | 0.260 | 0.520 |
| Other sports activities (dynamic games, team sports, etc.) | 0.956 | 0.989 |
| Gym | 0.287 | 1.232 |
| I have not done any physical activity in the past week | 0.029 | 0.217 |
| **Physical activities performed during the pandemic** | | |
| A walk lasting at least 10 min | 0.162 | 1.335 |
| Cycling | 0.739 | 1.101 |
| Running or jogging | 0.031 | 0.599 |
| Home activity, fitness, or exercise | 0.012 | 0.601 |
| Physical activities performed online at home | 0.866 | 1.081 |
| Other sports activities (dynamic games, team sports, etc.) | 0.117 | 0.664 |
| Gym | 0.013 | 0.539 |

**Table 3.** *Cont.*

| Dependent Variables | Sig. | Exp (B) |
|---|---|---|
| I have not done any physical activity in the past week | 0.488 | 1.570 |
| Gender | 0.078 | 0.724 |
| Frequency of participation in sports activities during the pandemic | 0.000 | 3.220 |
| The degree of difficulty regarding the practice of sports during the pandemic compared to the previous period of normality | 0.000 | 4.250 |
| Constant | 0.304 | 2.867 |

The personal factors associated with participating in sports and physical activities were the most important predictors, because it is 3 times more likely that those who considered that it was not more difficult to practice sports in a pandemic were part of the students who, during the pandemic, frequently participated in sports activities, because exp (b) = 3.220. In addition, the same probability is 4 times higher among students who reported positively the practice of sports activities during the pandemic, compared to the previous period, because exp (b) = 4.250.

The adequacy of the logistic regression model is significant (Nagelkerke R Square = 0.304).

The relationship of association between the degree of difficulty in carrying out sports activities since the beginning of COVID-19 and the frequency of participation in sports and physical activities during the pandemic:

Description of variables: Two ordinal variables were used to test this association. The independent variable used in this calculation is the variable that shows the degree of difficulty perceived by students in practicing sports activities during the pandemic (Question: How difficult has it been for you to do physical sports activities since the beginning of COVID-19?), and the dependent variable is the one that indicates the frequency with which they participate in physical sports activities (Question: How often did you participate in physical activities and sports during the pandemic?). Both variables used in this test are qualitative variables. To test the existence of the association between the two variables mentioned above, we used the chi-square test. Its value (273,487) is statistically significant ($p < 0.01$) for 20 degrees of freedom. Therefore, there are significant differences between students who practiced a sport more frequently and those who practiced less frequently during the pandemic depending on the perceived difficulty for this specific period. Thus, we can say, with the same probability of 99%, that this association exists among the researched population, not only in the sample. Both the PHI value (0.572) and the Cramer's V value (0.286) are statistically significant for $p < 0.01$ and indicate an association between the two variables.

The hypothesis was confirmed by the results obtained, which show that students who participate in sports activities with a high frequency are those who consider that playing sports during the pandemic was very easy; more precisely, they do sports 3–5 times a week (55.9%), while students who stated that practicing sports during this period was very difficult rarely participate in such activities, with only 3.9% of them doing sports 3–5 times a week.

An interesting fact is that almost a third of those who said that it was very difficult to play sports during the pandemic do sports once or twice a week. A high percentage play sports only once or twice a week, even if it was easy (40%) or very easy (34.9%) for them to do so during a pandemic. In fact, among students who reported doing sports only once or twice a week, the difficulties due to this period did not seem to influence the frequency with which they chose to do so (Figure 1).

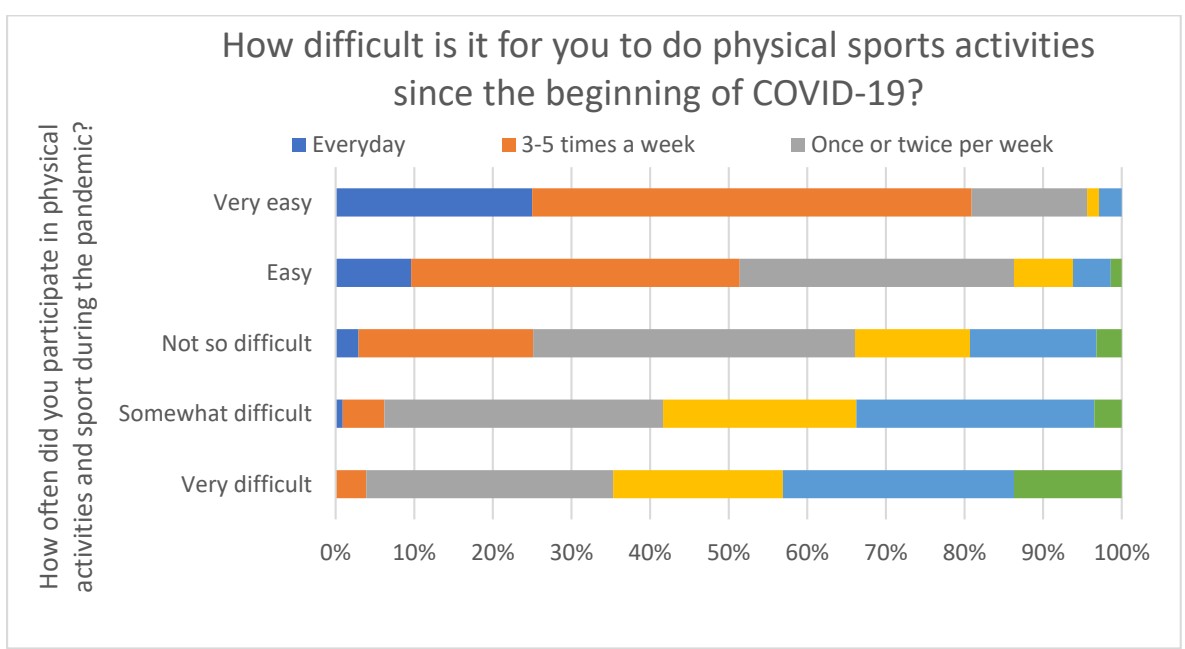

**Figure 1.** How often did you participate in physical activities and sports during the pandemic? How difficult has it been for you to do physical sports activities since the beginning of COVID-19?

Only three predictors proved to be more important in terms of the frequency with which students practiced physical activities and sports during the pandemic.

Thus, those who practiced sports regularly during the pandemic were almost twice as likely to be boys, because exp (b) = 1.759, almost three times more likely to be students who did not consider that it is more difficult to do sports during the pandemic, because exp (b) = 2.843, and almost twice as likely to be among the students who gave more importance to the practice of sports activities during the pandemic, because exp (b) = 2.350.

The adequacy of the logistic regression model is acceptable (Nagelkerke R Square = 0.226), as shown in Table 4.

**Table 4.** The results of the logistic regression regarding the statistical determinants of the probability that students would frequently engage in physical activities and sports during the pandemic.

| Dependent Variables | Sig. | Exp (B) |
|---|---|---|
| **Sports activity partners before the pandemic** | | |
| Family | 0.175 | 3.255 |
| Schoolmates | 0.400 | 2.011 |
| Friends | 0.234 | 2.453 |
| I do sport alone | 0.236 | 2.500 |
| **Sports activity partners during the pandemic** | | |
| Family | 0.442 | 0.553 |
| Schoolmates | 0.572 | 0.577 |
| Friends | 0.580 | 0.663 |
| I do sport alone | 0.446 | 0.569 |
| **Gender** | 0.001 | 1.759 |
| **The degree of difficulty regarding the practice of sports during the pandemic compared to the previous period of normality** | 0.000 | 2.843 |
| **How students relate to the practice of sports activities during the pandemic compared to the period before it** | 0.253 | 1.224 |
| **The importance given to sports activities since the introduction of the restrictions** | 0.000 | 2.350 |
| **The intention to do more physical activity and sports after the end of the restrictions** | 0.002 | 0.510 |
| **Constant** | 0.204 | 0.224 |

Figure 2 shows the major differences considering the partners with whom students usually participated in PA before and during the pandemic. Before COVID-19, 49% practiced sports with friends, 32.1% alone, and only 3.8% with family. The restrictions and the fear among the population have had a great impact on this aspect. During COVID-19, only 24.4% practiced PA with friends, 47.9% alone, and 11% with their families. These results show the major effects that this pandemic has had on the social part of their lives.

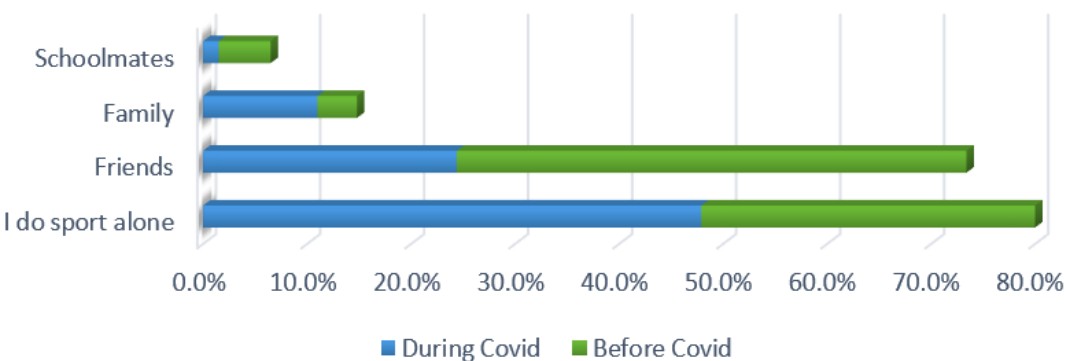

**Figure 2.** The partners with whom students usually participated in physical sports activities before/during the pandemic.

The most common locations for practicing PA and sports before the pandemic were the gym (33.6%), public spaces (23.6%), and outside (20.7%). Only 13.2% were doing sports at home. Due to fear, restrictions, and promoting of social distancing and staying home, 49.2% of the students started practicing PA in the house, while the number of them going to the gym (7.1%) or in public spaces (10.4%) has considerably dropped (Figure 3).

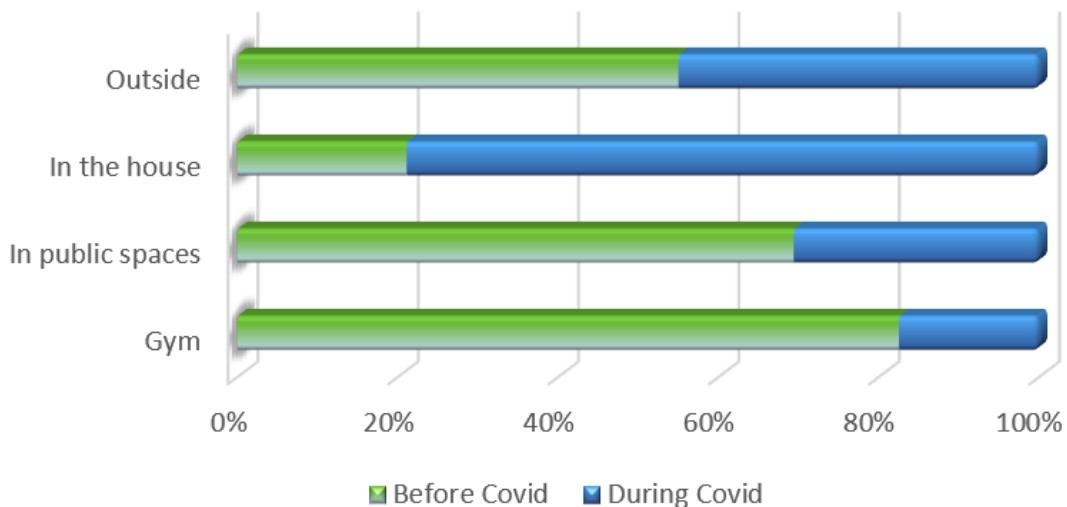

**Figure 3.** The most common location for physical activity and sports before/during COVID-19.

## 4. Discussion

Even though it is well known and documented that PA highly contributes to preventing health issues and reducing mortality risk factors while sedentarism increases the risk factors of heart diseases, anxiety levels, and obesity, because of the temporary extended closure of universities and learning institutions during the COVID-19 pandemic, the physical activity of students of all ages was dramatically reduced. In our study, it has been found that overall, the restrictions and lockdown as a measure against the spread of the COVID-19 pandemic have prevented students from complying with the WHO recommendation of moderate-to-vigorous physical activity during the period of home confinement [5,21].

Another important finding is that sport has been hit very hard during the pandemic, in such a way that has never been seen before. Face-to-face and group sports, but also going to the gym and other public spaces, were often turned into home workouts with family or alone during the period in which restrictions were imposed [22,23].

Our studies have also shown that many students that were engaged in PA and sports before the pandemic kept doing this also during it, changing only the places where they were going and their PA partners (Figure 1) [24]. However, the frequency of participating in sports activities dropped significantly during the pandemic.

Australian, Chinese, Italian, and Belgian students also demonstrated a rather negative change during lockdown [5,25–28]. It is however difficult to compare our data with theirs given the fact that their sample populations were not necessarily nationally representative, and the results of any country depend on different sampling methods, statistical modeling, and other tools [5].

Moreover, only 10% of university students met the guidelines for physical activity, while 30% met the guidelines for sedentary behavior during the pandemic. The minutes per week spent engaging in moderate-to-vigorous physical activity during the pandemic decreased by approximately 20% ($p < 0.001$), and the hours spent in sedentary activities increased by 3 h per day ($p < 0.001$) [5,24,29].

Australian, Chinese, Italian, and Belgian students also demonstrated a rather negative change during lockdown [5,25–28,30]. It is however difficult to compare our data with theirs given the fact that their sample populations were not necessarily nationally representative, and the results of any country depend on different sampling methods, statistical modeling, and other tools [5].

The WHO recommends practicing at least 150 min per week of moderate-to-vigorous-intensity physical activity or 75 min of high-intensity per week, or a combination of both [7,31,32].

In the European Union and Romania, ~60.0% and 45.0% of people, respectively, said they exercise or practice sports regularly or even seldomly [10]. In the present study, the majority that used to practice sports before the pandemic also found a way of doing so during it. The major difference was in fact the social aspect of doing sports, which plays a crucial role in their mental health. While many of them practiced sports with friends before the pandemic, they were forced to change this aspect of their habit. They started practicing PA alone, and this contributed to a lack of motivation for many of them. Alongside all these findings, 81% of people in Romania said that they have never or seldom done exercise or played sport—35% of them are men and 60% are women (all between 15 and 24 years old) [10].

In March 2020, immediately after the Romanian authorities applied lockdown, a new radical reality was encountered by the population. Given the great amount of free time and the necessity of finding new home activities, people started engaging in more physical activities. Similarly, other authors, such as Barkley et al., believe that the closure of sports facilities during the pandemic had a major impact on active students, leading to a decrease in their level of physical activity. In contrast, students who were less active before the pandemic significantly increased their overall level of physical activity [31,33].

The negative influence on the psychological well-being of people because of the reduction of PA is a fact; however, only two studies, conducted in Italy and Canada, have

shown that psychophysical conditions have worsened during the lockdown period of COVID-19. The lack of possibility for individuals to choose whether to perform or not physical activity hit them very hard. The closure of gyms, stadiums, pools, fitness studios, parks, etc. has compelled people to find different alternatives for exercising [6,31]. Online courses and the high level of use of the laptop were important factors that determined how students changed their daily routines.

Results have also shown that students are more likely to practice PA alone or in small groups of people because of the fear of getting sick. It is very important for universities to start implementing ways to monitor students' well-being and for them to receive psychological services. Unfortunately, there is no standardized measure that could be used to measure and evaluate students' mental health and to help them through the process. Using questions from the survey designed by the NCAA, which created a friendly measure, the SAWS (Student-Athlete Well-Being Scale), can ensured that students are being monitored for symptomatology of psychological distress. Moreover, the SAWS can help monitor treatment progress and all the outcomes of students using mental health services. We, along with many authors, emphasize the importance of practicing physical activity and combating a sedentary lifestyle, further suggesting the existence of another pandemic, namely, that of lack of physical activity [34–38].

Exercising regularly is a tool for combating a sedentary lifestyle and for maintaining a good health level. It is also known that PA can help overcome the health consequences of the COVID-19 pandemic in more than one way. For those with different health conditions (such as diabetes, cardiovascular disease, hypertension, and cancer), practicing PA regularly is a good way of recovering after having the virus [6,36,39].

In the studies identified for our research, we have found evidence for decreased physical activity during the lockdown, mainly in organized sports, as structured sports programs and facilities were closed. Public recommendations for an active lifestyle have been ineffective. An increase has been detected in outdoor play and unorganized activities [40–46], and this is related to lower levels of national restrictions [45]. The increase in outdoor activities depends strongly on the housing environment. The pandemic and its restrictions have further increased social differences between families with safe and spacious outdoor spaces.

As other important studies also show, students' motivation is influenced by various factors, such as intrinsic and identified motivation, introjected motivation, extrinsic motivation, and amotivation [47]. This is caused by many external factors but can be combated by PE teachers and by PA being carried out in the universities. It has been highlighted that PE teachers have a very important role in motivating students' participation in sport activities and in helping reduce sedentarism [48,49].

## 5. Conclusions

The objective of the article was to determine the impact of the COVID-19 pandemic on the well-being and physical health of the students.

Our results have shown that quarantine in Romania led to a significant increase in sedentarism among students and a reduction in the energy for physical activity, which also affects their psychological well-being. There has been detected a decrease in the levels of PA in free and leisure time because of the closure of sports facilities and all the restrictions imposed by the authorities. Therefore, our research confirms the hypothesis by highlighting the importance of promoting the practice of PA in our university, and in all universities, and also the fact that before the pandemic students were more eager to practice PA and were more motivated by the possibility of practicing it with their friends or families. Because of all the restrictions imposed, during the pandemic they had a lack of motivation and were less physically active than before. Students are willing to practice sports with their friends and family, take part in different activities, and remain physically active. Programs that promote the practice of sports and physical activities are necessary at educational authorities and political institutions, but also at sports clubs and by PE teachers to stimulate the participation of students during their free time.

Based on our results, we can confirm our hypothesis, stating the fact that the pandemic has had an important impact on the students' practice of physical activity and on the social interaction that sports bring. The restrictions that came with the pandemic have influenced their habits and decreased their motivation for being physically active. Participating in physical activities/different sports can improve motivation and can help students overcome the difficulties and differences that arose with the pandemic.

## 6. Limitations

Our findings must be interpreted in the light of some limitations. The questionnaires may be a reliable source of information regarding the practice of PA before and during the pandemic among students, but due to the current situation, we only addressed the study through online platforms and the internet.

The second limitation is regarding the sudden outbreak of COVID-19. We could not assess PA before and during the pandemic because of the rigorous local restrictions [31].

Another important limitation of this study is that we have data of the consequences of only a few months of the total lockdown period [50,51]. We have tried to reach as many students as we could via any available social media tools so that the responses to the questionnaire were representative, but not all students are characteristic of the adult and Romanian population.

As infection rates and restrictions differed, our findings cannot be related to specific lockdown measures. However, we tried to take these differences into account by considering the national context when interpreting studies.

However, we strongly believe that this study is a significant one in regard to demonstrating the impact of COVID-19 on students, and in analyzing how their day-by-day exercising has changed [52].

**Author Contributions:** All authors contributed equally to the project conceptualization; methodology; resources and validation; software; formal analysis; investigation; data curation; writing—original draft preparation; writing—review and editing; visualization; supervision; project administration; funding acquisition. All authors have read and agreed to the published version of the manuscript.

**Funding:** This research received no external funding.

**Institutional Review Board Statement:** The study was conducted in accordance with the Declaration of Helsinki, and approved by the Institutional Ethics Committee of Technical University of Cluj-Napoca (protocol code 414, 17/03/2022).

**Informed Consent Statement:** Informed consent was obtained from all subjects involved in the study.

**Acknowledgments:** The authors would like to thank all the volunteers for their participation in this research project and for kindly sharing their personal information.

**Conflicts of Interest:** The authors declare no conflict of interest.

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
