# Peer review of "COVID-19 Pandemic: Impediment or Opportunity? Considerations Regarding the Physical-Health Impact and Well-Being among Romanian University Students"

_applsci, doi:10.3390/app12188944_

Round 1
Reviewer 1 Report
Abstract
Abbreviations are used, but it is not stated what they mean (PA and PE). The full name should be given before using abbreviations.
If there is no other gender category besides F and M (and in this research there is none), then it is enough to indicate how many were of one gender (e.g., 57.73% male). It goes without saying that the rest is female.
2. Materials and Methods
pp. 3, line 109 – What are the „busy programs“?
pp. 3, line 124 – „a questionnaire based on questions from several questionnaires“ – which questionnaires? It would be beneficent if at least the references were written.
3. Results
The first sentence in the results section lists some "results". But it is not clear what was calculated, which test was applied. Is it perhaps just a descriptive data comment? Is it perhaps a t-test comparing PA before and during the pandemic? It is necessary to clearly state what was applied and for what purpose, and only then state the result.
pp. 7, line 218-220 – the hypothesis is stated. It is not clear where this hypothesis came from? On what basis was it built or set up? In addition, it is unusual for the hypothesis to appear (for the first time) in the results section of the paper.
Conclusion
"Based on our results, we confirm our hypothesis, stating the fact that physical activity has a big impact on health, mental health, and socializing." - I would suggest softening this statement considering that you have not established all of the above. This conclusion goes beyond the scope of this research.
Author Response
Dear reviewer,
First and foremost, thank you very much for your time and valuable comments, which all have been considered and incorporated. The detailed list of responses is given below. We hope that the modifications and explanation will be acceptable to you.
Yours sincerely,
Adrian Suciu, corresponding author
Abbreviations are used, but it is not stated what they mean (PA and PE). The full name should be given before using abbreviations.
If there is no other gender category besides F and M (and in this research there is none), then it is enough to indicate how many were of one gender (e.g., 57.73% male). It goes without saying that the rest is female
A: Thank you very much for your comment. This has been all modified and can be seen in the revised version.
- 3, line 109 – What are the „busy programs“?
- 3, line 124 – „a questionnaire based on questions from several questionnaires“ – which questionnaires? It would be beneficent if at least the references were written.
A: Thank you very much for your suggestion. We have explained what busy programs are.
Regarding the questionnaires, the references have been added. Thank you very much for your observation.
The first sentence in the results section lists some "results". But it is not clear what was calculated, which test was applied. Is it perhaps just a descriptive data comment? Is it perhaps a t-test comparing PA before and during the pandemic? It is necessary to clearly state what was applied and for what purpose, and only then state the result.
- 7, line 218-220 – the hypothesis is stated. It is not clear where this hypothesis came from? On what basis was it built or set up? In addition, it is unusual for the hypothesis to appear (for the first time) in the results section of the paper.
A: Thank you very much for your comment. This has been clarified in the revised version.
Also, the hypotheses has been added in the introduction and we hope it is clearer now.
"Based on our results, we confirm our hypothesis, stating the fact that physical activity has a big impact on health, mental health, and socializing." - I would suggest softening this statement considering that you have not established all of the above. This conclusion goes beyond the scope of this research.
A: Thank you very much for this suggestion!
This has been modified.
Reviewer 2 Report
The abstract introduces the content of the paper appropriately. The introduced theories and previous research results are in line with the content and direction of the paper. However, besides the aims, basic research questions or exact hypotheses should be introduced clearly, possible at the end of the Introduction section. Now the small section entitled Hypotheses is rather a brief summary of the aims. The hypotheses introduced later in the Results section should be placed here.
Concerning the Methods section, the period and place of the research, the sample, the tests and the statistical processes are introduced. It would be better to change the titles in some cases (Subjects to 'Sample', Applied tests to 'Instruments'). Also, it would be necessary to introduce the content of the questionnaire in detail. The sampling method should be introduced in the 'Sample' section instead of the 'Instruments'.
The Results section should be improved. First, we can read about the results of the regression analysis; however, its role concerning the hypotheses is not clear. Also, it would be more understandable to read the results of the descriptive statistics as we usually start with the 'easier' analysis. Figures should be unique (now the authors use different styles).
Discussion and conclusions are almost correctly defined. It should be necessary to reflect on the assumptions of the authors i.e. the research questions or hypotheses. After reviewing the hypotheses, the conclusions should be modified as well. The authors try to compare data concerning the times before and after the outbreak of the pandemic. I also suggest some relevant literature adding to the paper:
Kovács, K.; Kovács, K.E. Using the Behavioural Regulation in an Exercise Questionnaire (BREQ–2) in Central and Eastern Europe: Evidence of Reliability, Sociocultural Background, and the Effect on Sports Activity. Int. J. Environ. Res. Public Health 2021, 18, 11834. https://doi.org/10.3390/ijerph182211834
Iglesias-Martínez, E.; Roces-Garcia, J.; Méndez-Alonso, D. Predictive Strength of Contextual and Personal Variables in Soccer Players’ Goal Orientations. Int. J. Environ. Res. Public Health 2021, 18, 9401. https://doi.org/10.3390/ijerph18179401”
Bollók, S.; Takács, J.; Kalmár, Z.; Dobay, B. External and Internal Sport Motivations of Young Adults. Biomed. Hum. Kinet. 2011, 3, 101–105.
In my opinion, this paper is an intriguing paper introducing extremely important practical information. After the modifications, it is worth publishing this report.
Author Response
Dear reviewer,
First and foremost, thank you very much for your time and valuable comments, which all have been considered and incorporated. The detailed list of responses is given below. We hope that the modifications and explanation will be acceptable to you. All the modifications can be seen in the revised manuscript.
Yours sincerely,
Adrian Suciu, corresponding author
The abstract introduces the content of the paper appropriately. The introduced theories and previous research results are in line with the content and direction of the paper. However, besides the aims, basic research questions or exact hypotheses should be introduced clearly, possible at the end of the Introduction section. Now the small section entitled Hypotheses is rather a brief summary of the aims. The hypotheses introduced later in the Results section should be placed here.
A: Thank you very much for your comment. The hypotheses have been placed in the introduction, so it is clearer what results we expect.
Concerning the Methods section, the period and place of the research, the sample, the tests and the statistical processes are introduced. It would be better to change the titles in some cases (Subjects to 'Sample', Applied tests to 'Instruments'). Also, it would be necessary to introduce the content of the questionnaire in detail. The sampling method should be introduced in the 'Sample' section instead of the 'Instruments'.
A: Thank you very much for your suggestion. We have modified this.
The Results section should be improved. First, we can read about the results of the regression analysis; however, its role concerning the hypotheses is not clear. Also, it would be more understandable to read the results of the descriptive statistics as we usually start with the 'easier' analysis. Figures should be unique (now the authors use different styles).
A: Thank you for your comment! The results of the regression analysis is very important, as it shows that students considered it more difficult to practice sports activities during the pandemic, and also shows their reasons in doing so. This was very important in order to determine what is their main motivation in being physically active.
Discussion and conclusions are almost correctly defined. It should be necessary to reflect on the assumptions of the authors i.e. the research questions or hypotheses. After reviewing the hypotheses, the conclusions should be modified as well. The authors try to compare data concerning the times before and after the outbreak of the pandemic. I also suggest some relevant literature adding to the paper
A: Your suggestions were very helpful, so we have added the recommended literature to our article. The Conclusion has also been modified.
Reviewer 3 Report
The main aim of this study was to determine if and how the pandemic affected the university students’ mental and physical health in Romania. Regarding the authors, I would like to thank them for their effort and motivation involved in this research study. Unfortunately, the article does not add anything new to the science of mental and physical health in the COVID-19 pandemic. It is simply another example confirming previously discovered aspects. It may be the first such study in the case of Romanian students, but the results obtained are the same as other publications on student health in other countries, which were also published in the journals of the MDPI publishing house back in 2020. Due to the negligible contribution of the presented content to current science, I do not see the possibility of publishing the manuscript in Applied Sciences. It may be worthwhile for the authors to send their article to a journal with a more local focus, potentially resulting in a higher readership. Before that, however, the discussion should be strengthened and related to other similar studies, and the introduction should be significantly expanded, without duplicating the repeatedly cited data on WHO or typical behaviors in the COVID-19 pandemic. I keep my fingers crossed for the final success of publishing the manuscript in another journal.
Author Response
Dear reviewer,
First and foremost, thank you for your patience in reading our study and for your remarks.
This journal paper has a clear theme, and its research purpose was to determine the impact of the COVID-19 pandemic on the Romanian university students’ well-being and physical health during quarantine, which is a very important aspect for our country. While the sedentary level is very high in Romania among young adults and students, it was important to determine how and if the pandemic affected this level. Being a Technical University, Physical Education and Sports are not a priority among students. That is why we wanted to see how we could motivate them by knowing different factors that determine them being more physically active.
This issue (Smart Education through Physical Activity and Sport) is also about how we could use PE and Sports to help students learn more efficient and have a healthy lifestyle also when they are young.
We find this study relevant for this number, regarding the important number of students who answered the questionnaire and offered us a better view on how we can adapt PE classes after and during the pandemic, considering that there are still some imposed restrictions regarding PA classes in universities in Romania.
Regarding your recommendation to include other studies, we have done this, and you can see the latest version of our manuscript in the attachment.
Thank you again for your patience and recommendations!
Round 2
Reviewer 2 Report
The author corrected the paper following my previous suggestions, the modifications and answers are correct. Congratulation to the authors!
Author Response
Dear reviewer,
We want to thank you very much for all your valuable comments and remarks regarding our article.
Yours sincerely,
Adrian Suciu, corresponding author
Reviewer 3 Report
Thank you for making the corrections. Although I still have some concerns and think that the manuscript does not quite fit this journal, I appreciate your efforts to improve the content of the article as much as possible. Please add information on whether the research was conducted in accordance with the Declaration of Helsinki regarding anonymity and obtaining consents. This is extremely important information that implicates the proper conduct of the research. Good luck in your further projects!
Author Response
Dear reviewer,
We would like to thank you again for all your time and remarks regarding our article.
Yes, we have added that this article was conducted in accordance with the Declaration of Helsinki regarding anonymity and obtaining consents. Also we have the Ethics-comitee-approval from our university. We have added this in the manuscript.
Thank you very much for your observation.
Yours sincerely,
Adrian Suciu, corresponding author